# Artificial Light at Night, Sleep Disruption, and Liver Health: Implications for MASLD Pathogenesis

**DOI:** 10.3390/ijerph22111729

**Published:** 2025-11-15

**Authors:** Damaris Guadalupe Nieva-Ramírez, Misael Uribe, Natalia Nuño-Lámbarri

**Affiliations:** 1Traslational Research Unit, Medica Sur Clinic & Foundation, Puente de Piedra 150, Toriello Guerra Tlalpan, Mexico City 14050, Mexico; damaris.nievar@alumno.buap.mx; 2Obesity and Digestive Diseases Unit, Medica Sur Clinic & Foundation, Puente de Piedra 150, Toriello Guerra Tlalpan, Mexico City 14050, Mexico; muribe@medicasur.org.mx; 3Department of Surgery, Faculty of Medicine, The National Autonomous University of Mexico (UNAM), Mexico City 14050, Mexico

**Keywords:** artificial light at night, circadian disruption, melatonin suppression, MASLD, light pollution, hepatic lipid metabolism

## Abstract

This narrative review explores the role of artificial light at night (ALAN) as an emerging environmental determinant of liver and metabolic health, with particular emphasis on its contribution to metabolic dysfunction-associated steatotic liver disease (MASLD). The objective was to synthesize and critically evaluate experimental and epidemiological evidence linking nocturnal light exposure, circadian disruption, and hepatic metabolic alterations. Literature was retrieved from PubMed, Scopus, Web of Science, and Google Scholar databases covering the period 1980–2025 using predefined search terms related to ALAN, circadian rhythm disturbance, melatonin suppression, sleep disruption, and MASLD. Relevant experimental studies in animal models and observational studies in humans were included. Evidence indicates that blue-enriched light (~460–480 nm) suppresses melatonin, desynchronizes central and hepatic circadian clocks, and disrupts glucose–lipid metabolism, leading to insulin resistance, oxidative stress, and hepatic steatosis. Chronic ALAN exposure also alters gut microbiota composition and increases intestinal permeability, suggesting a light–gut–liver axis in MASLD pathogenesis. Human epidemiological studies associate higher environmental ALAN exposure with obesity, metabolic syndrome, and poor sleep quality—recognized risk factors for MASLD. Recognizing ALAN as a modifiable environmental exposure highlights the need for public health strategies and clinical guidelines to mitigate its metabolic impact through improved lighting design and sleep hygiene.

## 1. Introduction

Over the past century, the widespread adoption of electric lighting has markedly increased human exposure to artificial light at night (ALAN), raising concerns about its impact on human physiology. ALAN is a potent modulator of circadian rhythms—endogenous, approximately 24 h cycles that regulate essential physiological processes such as sleep–wake patterns, core body temperature, hormone secretion, blood pressure, and energy metabolism. These rhythms are orchestrated by the central biological clock, located in the suprachiasmatic nucleus (SCN) of the hypothalamus, which receives direct photic input from intrinsically photosensitive retinal ganglion cells (ipRGCs). Light signals from ipRGCs synchronize both the SCN and peripheral oscillators present in metabolically relevant tissues, including the liver, pancreas, and adipose tissue [1,2,3]. Proper synchronization of these molecular clocks is essential for metabolic balance, whereas circadian disruption—whether caused by shift work, jet lag, sleep restriction, or prolonged ALAN exposure—has been associated with significant metabolic disturbances [4,5,6].

Chronic exposure to ALAN, particularly during the night, has been shown to exacerbate insulin resistance, systemic inflammation, and hepatic dysfunction, especially when combined with high-fat diets [7,8,9]. These alterations align closely with the pathophysiological features of metabolic dysfunction-associated steatotic liver disease (MASLD) [10]. MASLD is characterized by excessive hepatic fat accumulation in the absence of significant alcohol intake and is driven by multiple, interrelated mechanisms, including insulin resistance, mitochondrial dysfunction, oxidative stress, chronic inflammation, and genetic susceptibility [9,10]. Its global prevalence is alarmingly high, affecting approximately 38.77% of the general population [11] and up to 50.7% of individuals with overweight or obesity [12], with particularly elevated rates in East Asia and the Middle East [13].

Given that ALAN-induced circadian disruption can perturb lipid metabolism, insulin sensitivity, and gut microbiota composition—processes central to MASLD pathogenesis—it is plausible that light pollution may play a significant role in both the onset and progression of this condition. Therefore, this review aims to synthesize current scientific evidence on the relationship between ALAN exposure, including the influence of specific light wavelengths, and the pathophysiology of MASLD, with an emphasis on circadian rhythm–mediated metabolic mechanisms.

While previous reviews have independently explored the metabolic effects of circadian disruption or the systemic consequences of artificial light exposure, few have specifically addressed how nocturnal light environments contribute to hepatic lipid dysregulation and the pathogenesis of metabolic dysfunction-associated steatotic liver disease (MASLD). Moreover, existing studies often focus on either experimental animal models or isolated endocrine outcomes without integrating these findings into a unified hepatic framework. This review aims to bridge that gap by synthesizing multidisciplinary evidence—from chronobiology, hepatology, and environmental health—to delineate the mechanistic pathways through which ALAN influences hepatic metabolism, circadian gene expression, and gut–liver interactions. By clarifying these links, this paper contributes to expanding current understanding of MASLD as a multifactorial disease influenced not only by diet and lifestyle but also by modifiable environmental light exposure.

## 2. Methodology

A comprehensive literature search was performed across major scientific databases including PubMed/MEDLINE, Scopus, Web of Science, and Google Scholar, covering publications from 1980 to 2025. The search strategy combined controlled vocabulary and free-text terms related to artificial light at night (ALAN), circadian disruption, sleep disturbance, melatonin suppression, and metabolic dysfunction-associated steatotic liver disease (MASLD), using Boolean operators (AND, OR) to broaden and refine results. Inclusion criteria comprised original experimental and observational studies published in English that examined the effects of ALAN or circadian disruption on hepatic, metabolic, or sleep-related outcomes. Both animal and human studies were considered if they reported measurable endpoints related to liver function, lipid metabolism, insulin resistance, or melatonin regulation. Exclusion criteria included editorials, conference abstracts, studies with insufficient methodological details, and those unrelated to hepatic or metabolic endpoints. Artificial intelligence tools (ChatGPT) were employed solely to assist with English language refinement, including grammar, syntax, and style improvements. No content, data, or scientific interpretation was generated or modified by these tools. 

## 3. Artificial Light at Night: Characteristics, Exposure Patterns, and Biological Implications

### 3.1. Spectral Characteristics, Sources, and Health Implications

Artificial light at night (ALAN) has become an omnipresent component of modern environments, affecting both urban and rural areas. While the widespread use of electric lighting has improved safety, productivity, and quality of life, it has also raised growing concerns about its unintended consequences on human health and ecosystems [14]. The potential biological impact of ALAN is determined not only by its intensity—quantified as the number of photons per unit area—but also by its spectral composition, since different wavelengths can exert highly specific physiological effects [15,16].

From a methodological perspective, the most accurate approach for assessing ALAN involves physical photometric units such as photons·m^−2^·s^−1^, complemented by spectral analysis to characterize wavelength distribution. However, in practice, most studies still rely on lux or footcandle measurements, which are weighted according to human visual sensitivity and therefore fail to fully reflect the biological relevance of light exposure [16].

Different ALAN sources present distinct emission spectra and usage contexts. Incandescent lamps emit a continuous spectrum with a higher proportion of longer wavelengths (600–800 nm), resulting in relatively lower melatonin suppression. Despite this, they are energetically inefficient and have been largely replaced by other technologies, except in some rural areas or in decorative and historical lighting [17]. Fluorescent lamps generate light by exciting phosphors with ultraviolet radiation and produce spectral peaks between 450 and 600 nm, including blue light—well known to disrupt circadian rhythms [15]. Although fluorescent lamps have traditionally been common in hospitals, schools, 24 h service stations, commercial offices and other interior applications they are increasingly being replaced by energy-efficient LED systems, particularly in high-income countries such as the United States, where large-scale retrofitting projects have accelerated this transition over the past decade. LED replacements are also common in exterior street and highway lighting, further contributing to the global shift toward solid-state illumination technologies. Sodium vapor lamps, extensively used in street lighting and industrial complexes, emit predominantly in the yellow range (~589 nm); low-pressure types produce nearly monochromatic light, while high-pressure variants have a broader spectrum but still exhibit a relatively low circadian impact [16]. In contrast, light-emitting diodes (LEDs), especially cool-white types, emit strongly in the blue range (460–480 nm). While highly energy-efficient and long-lasting, LEDs can strongly suppress melatonin and disrupt circadian regulation due to their spectral composition. Notably, even LEDs with the same correlated color temperature may differ substantially in their melanopic spectral impact, resulting in variable biological effects despite similar visual appearance [7,18].

These technologies are integrated into a variety of real-world applications, ranging from public streetlights and residential illumination to architectural lighting, commercial displays, advertising billboards, and the screens of personal electronic devices. Remote sensing studies reveal that urban skyglow—the diffuse brightening of the night sky—is primarily generated by public and domestic lighting sources [19].

However, the difference between an incandescent light source and a blue-rich daylight spectrum is relatively minor compared to the influence of illuminance level on visual and non-visual responses. When light intensity is held constant, changing the spectral composition—from warm, low-color-temperature incandescent light to a blue-enriched daylight spectrum—produces only modest physiological or perceptual effects. In contrast, variations in illuminance (brightness in lux) have a far greater impact on alertness, circadian regulation, and overall visual perception. This finding highlight that in practical lighting design and human performance research, the quantity of light often outweighs its spectral composition, emphasizing the need to prioritize appropriate light levels rather than relying solely on blue-enriched illumination to achieve desired biological or cognitive outcomes [20].

The health implications of chronic ALAN exposure are significant, especially when the emitted light is enriched in blue wavelengths. Nighttime exposure to such light has been consistently associated with suppression of nocturnal melatonin secretion, an effect that extends beyond sleep disruption to include alterations in mood, endocrine signaling, metabolic homeostasis, and cardiovascular health [21,22,23]. These effects are not merely theoretical: large-scale research in China has demonstrated that individuals living in areas with higher environmental ALAN levels have a significantly greater prevalence of insomnia [24]. Similarly, in a large cross-sectional study of U.S. adolescents, higher outdoor ALAN exposure was associated with measurable disruptions in sleep patterns and an increased prevalence of mental health disorders. Adolescents living in high-exposure areas reported later weeknight bedtimes—approximately 29 min later in the highest compared with the lowest exposure quartile—and shorter sleep durations of about 11 min, even after adjusting for sociodemographic and area-level variables. Moreover, elevated ALAN levels were linked to greater odds of mood and anxiety disorders, with particularly strong associations for bipolar disorder (odds ratio ≈ 1.19) and specific phobias (odds ratio ≈ 1.18). Collectively, these findings underscore that outdoor ALAN functions as an important environmental determinant of both circadian regulation and psychological health, highlighting the need to consider community-level lighting environments alongside individual light exposure when evaluating the broader health consequences of nocturnal illumination [25].

Contrary, a recent field study investigated the effects of solid-state roadway lighting on human melatonin suppression under realistic nighttime conditions. Participants, including drivers, pedestrians, and individuals exposed to light trespass in bedrooms, were exposed to five types of LED lighting with different spectral compositions, as well as a no-lighting control condition. This study found no significant differences in salivary melatonin concentrations between lighting conditions, regardless of whether the spectrum was blue-enriched or contained lower short-wavelength content. These findings suggest that, at typical roadway lighting intensities, spectral composition has a negligible effect on melatonin suppression. Consequently, standard LED-based roadway lighting, when properly designed and implemented, is unlikely to disrupt human circadian rhythms or compromise nocturnal physiological regulation [26].

### 3.2. Influence of Intensity, Duration, and Spectral Composition

The specific metabolic and endocrine effects of ALAN cannot be fully understood without considering its intensity, duration of exposure, and spectral composition.

Even very low illuminance levels can trigger measurable biological changes. In hypertensive rat models, exposure to 1–2 lux for 2–5 weeks led to sustained increases in blood pressure, insulin resistance, and hepatic triglyceride accumulation [27]. In TALLYHO/JngJ mice, a strain predisposed to diabetes, 5 lux exposure for four weeks induced weight gain, hyperglycemia, and disruption of feeding rhythms. Interestingly, these effects were reversible after returning the animals to dark conditions, highlighting the dynamic adaptability of the circadian-metabolic axis. Higher intensities, such as 40 lux or more—comparable to the brightness of typical electronic screens—further exacerbate metabolic dysfunction, promoting hepatic steatosis and insulin resistance in susceptible models [28].

The duration of exposure is equally critical. Chronic ALAN exposure lasting four weeks or more in rodent models has been shown to induce persistent circadian disruption, hepatic inflammation, and hypertriglyceridemia [27]. In humans, even intermittent exposure, such as sleeping with a dim nightlight, has been linked to an increased risk of obesity and metabolic syndrome, both recognized risk factors for MASLD [29].

Spectral composition further modulates ALAN’s biological effects. Blue light around 480 nm is the most potent suppressor of melatonin and a major disruptor of circadian rhythms, with documented consequences on hepatic clock gene expression and mitochondrial function [30]. Cold-white LEDs, rich in blue wavelengths, have been shown to aggravate adipocyte hypertrophy and promote hepatic and adipose tissue dysfunction in animal models [8]. Notably, wavelength-specific effects have been observed: blue light exposure increases Per1 expression and lowers plasma glucose, while green light at 525 nm induces Per2 overexpression, indicating that ALAN spectral quality modulates hepatic metabolism in distinct ways [31]. Prolonged exposure to blue LED light over 44 weeks has been shown to induce hepatic lipid accumulation, reduce Kupffer cell density, and alter the expression of Bmal1 and Rev-erbα, even in the absence of high-fat dietary triggers, suggesting that light alone can initiate early MASLD-like changes [32]. In dietary interaction studies, green light combined with a high-fat diet resulted in more severe hepatic steatosis and pancreatic dysfunction than blue or white light, indicating that ambient lighting can modulate the metabolic impact of diet [33]. Furthermore, continuous light exposure, effectively eliminating nighttime darkness, accelerates steatohepatitis progression, partly through gut microbiota dysbiosis, impaired intestinal barrier integrity, and increased hepatic inflammation [34].

## 4. Light-Driven Regulation of the Circadian System

The circadian rhythm is an endogenous biological process that follows an approximately 24 h cycle and plays a fundamental role in regulating a wide range of physiological functions, including the sleep–wake cycle, body temperature, hormone secretion, and energy metabolism [3,35]. In mammals, the master circadian pacemaker is in the SCN of the hypothalamus. This structure is responsible for synchronizing internal biological rhythms with the external light–dark cycle, ensuring that physiological processes occur at the most appropriate times of day.

Light is the primary environmental cue, or zeitgeber, for circadian synchronization. The SCN receives photic input directly from the retina through a specialized subset of retinal ganglion cells known as ipRGCs. These cells express the photopigment melanopsin, which confers intrinsic light sensitivity, particularly to short-wavelength blue light in the range of approximately 460–480 nm [1,36]. Signals from ipRGCs travel via the retinohypothalamic tract to the SCN, where neurotransmitters such as glutamate and pituitary adenylate cyclase-activating polypeptide mediate entrainment of the central clock.

Within the SCN, photic stimulation regulates the transcription of core clock genes. The transcription factors Circadian Locomotor Output Cycles Kaput (CLOCK) and Brain and Muscle ARNT-Like 1 (BMAL1) form heterodimers that bind to E-box elements in DNA, activating the transcription of Period (Per1, Per2) and Cryptochrome (Cry1, Cry2) genes. The translated PER and CRY proteins accumulate in the cytoplasm, form complexes, and then translocate back to the nucleus, where they inhibit CLOCK:BMAL1 activity. This negative feedback loop generates self-sustaining circadian oscillations with a periodicity of approximately 24 h [37].

The photoresponse of ipRGCs is characterized by slow depolarization and requires relatively high light intensities, allowing them to integrate sustained light exposure while filtering out brief fluctuations in illumination [1]. Although ipRGCs are central to non-visual photoreception, they do not function in isolation; they also receive synaptic inputs from rods and cones, enabling integration of both image-forming and non-image-forming visual information. This integration underlies several physiological responses, including melatonin suppression, regulation of sleep, and control of the pupillary light reflex. At least five subtypes of ipRGCs (M1–M5) have been described, with M1 and M2 primarily projecting to the SCN, while other subtypes are involved in sleep regulation and pupillary responses [38].

Melatonin synthesis and secretion by the pineal gland are tightly regulated by the light–dark cycle. During daylight hours, light exposure inhibits melatonin production. In darkness, the enzyme arylalkylamine N-acetyltransferase becomes active, initiating melatonin biosynthesis [2,39]. Melatonin levels rise during the night, peaking in the early morning hours, and are markedly suppressed by exposure to light, especially in the blue wavelength range. This hormone acts as a systemic signal of darkness, circulating in both the bloodstream and cerebrospinal fluid. It influences a variety of physiological processes, including the suppression of neuronal activity in the SCN to promote sleep onset and adjustment of the circadian clock to environmental cues. Melatonin also modulates the expression of clock genes such as Per1 and Per2, contributing to the stability of circadian rhythms [40].

## 5. Circadian Disruption and Metabolic Dysfunction

Given the pivotal role of the circadian system in regulating metabolic homeostasis, its disruption has emerged as a critical mechanistic link between environmental factors and metabolic disease. The liver, as a central organ for energy balance, is particularly sensitive to alterations in circadian signaling and sleep architecture. ALAN and sleep loss can desynchronize central and peripheral clocks, impair hormonal regulation, and promote metabolic inflexibility. The following sections examine in detail how circadian misalignment, sleep disruption, and melatonin deficiency converge to disturb hepatic metabolism, leading to lipid accumulation, inflammation, and the progression of MASLD. (Figure 1).

### 5.1. From Molecular Mechanisms to Disease

Circadian disruption, defined as the misalignment between internal biological rhythms and the external light–dark cycle, has been strongly linked to a variety of adverse metabolic outcomes, including obesity, metabolic syndrome, and type 2 diabetes [5]. The underlying mechanisms involve alterations in the expression and function of core clock genes such as CLOCK, BMAL1, PER, and CRY, which play essential roles in regulating cellular metabolism. Dysregulation of these genes can lead to impaired glucose and lipid homeostasis, as evidenced in both animal experiments and human epidemiological studies [4,6,41].

Circadian misalignment affects the metabolic functions of multiple peripheral tissues. In the liver, disruption of the temporal coordination between feeding and endogenous metabolic cycles promotes dyslipidemia. In adipose tissue, circadian desynchrony impairs insulin sensitivity and reduces the capacity for lipid storage [42]. In skeletal muscle, misalignment can decrease glucose uptake efficiency, contributing to systemic insulin resistance.

Experimental evidence in animal models demonstrates that disruption of peripheral clocks, particularly in the liver and pancreas, interferes with the rhythmic expression of genes involved in carbohydrate and lipid metabolism. Mice with mutations in the CLOCK gene develop obesity, hyperglycemia, and hepatic steatosis, providing a direct mechanistic link between circadian disruption and metabolic syndrome [41]. Similarly, simulated jet lag conditions in mice lead to altered gut microbiota rhythmicity, impaired cellular metabolism, and increased susceptibility to obesity and diabetes [43].

In human studies, prolonged sleep restriction combined with circadian misalignment has been shown to decrease resting metabolic rate, impair insulin secretion, and induce a prediabetic state. In one controlled experiment, participants subjected to a 28 h light–dark cycle with only 5.6 h of sleep per 24 h for three weeks experienced reduced metabolic rate and diminished postprandial insulin response, leading to elevated blood glucose within the prediabetic range [4]. Epidemiological data further support these findings: shift workers, especially those working at night, have higher risks of obesity and type 2 diabetes. The Nurses’ Health Study II found that nurses working rotating night shifts had greater caloric intake and a higher incidence of obesity than those who never worked night shifts [6].

Exposure to ALAN during sleep has also been linked to increased obesity risk. Data from the U.S. Sister Study cohort indicated that women exposed to artificial light from sources such as televisions or lamps during sleep had a 17% higher likelihood of gaining at least 5 kg and a 33% increased risk of developing obesity over a five-year follow-up compared to those sleeping in complete darkness [44]. Similarly, studies in urban Chinese populations have found that residents of areas with higher levels of light pollution report poorer sleep quality and shorter sleep duration, even after adjusting for socioeconomic and lifestyle factors [45]. Higher nighttime light levels have also been associated with more frequent nighttime awakenings and reduced sleep efficiency, both of which are known to promote insulin resistance [46].

### 5.2. Impact of Sleep Disruption on Hepatic Metabolism and Its Role in MASLD Pathogenesis

Sleep plays a fundamental role in the regulation of hepatic metabolism through the coordinated action of circadian and hormonal mechanisms. Disruption of sleep—whether acute or chronic—has direct consequences on essential hepatic functions such as gluconeogenesis, lipogenesis, β-oxidation, detoxification, and xenobiotic metabolism. Experimental and clinical evidence shows that both short-term and prolonged sleep deprivation can induce metabolic changes in the liver that favor lipid accumulation, inflammation, and insulin resistance [47,48,49].

During periods of sleep loss, there is sustained activation of the sympathetic nervous system accompanied by elevated secretion of cortisol and catecholamines. These hormonal changes increase lipolysis in peripheral adipose tissue, raising the flux of free fatty acids (FFAs) to the liver. At the same time, the hepatic expression of genes involved in de novo lipogenesis, such as Elovl3, Lpin1, and Acot1, is upregulated, leading to the esterification of FFAs into triglycerides within hepatocytes. In the absence of a proportional increase in mitochondrial β-oxidation, these processes promote intrahepatocellular lipid accumulation, resulting in hepatic steatosis [48].

Sleep restriction also disrupts the synchronization between the central circadian pacemaker, located in the suprachiasmatic nucleus (SCN), and the liver’s own molecular clock. The hepatic clock, regulated by genes such as BMAL1, CLOCK, PER, and CRY, orchestrates the temporal expression of metabolic genes to align anabolic and catabolic processes with the sleep–wake cycle. Sleep deprivation alters the oscillatory patterns of these genes, causing metabolic desynchrony characterized by an overlap of anabolic and catabolic phases. This misalignment favors the accumulation of metabolic substrates such as glucose and lipids [50].

In addition to transcriptional changes, sleep disruption affects key metabolic cofactors. It reduces Nicotinamide adenine dinucleotide (NAD^+^) levels and increases the S-adenosylmethionine/S-adenosylhomocysteine (SAM/SAH) ratio, which is crucial for epigenetic methylation. This hypermethylation state can alter the expression of genes involved in lipid metabolism. Sleep loss also increases the levels of citric acid cycle intermediates such as citrate, which serves as a substrate for lipogenesis by activating acetyl-CoA carboxylase and upregulating ATP citrate lyase, thereby promoting the conversion of citrate to cytosolic acetyl-CoA and enhancing fatty acid synthesis [47,48,49].

Animal studies provide further mechanistic insights, showing that sleep deprivation alters hepatic energy metabolism pathways, including those involving glutathione, fructose and mannose metabolism, and pyruvate metabolism, potentially contributing to the development of metabolic disorders and obesity [48].

Human studies have identified similar associations between sleep duration and liver health. Both short and excessively long sleep durations have been linked to an increased risk of MASLD, cirrhosis, hepatocellular carcinoma, and liver-related mortality [51]. Pediatric studies also suggest a role for sleep in modulating hepatic fat content; for example, research has shown that sleep duration influences the relationship between body fat distribution and hepatic steatosis, indicating that improving sleep patterns may represent a modifiable target for managing fatty liver disease in children and adolescents [52].

### 5.3. Role of Melatonin Deficiency in ALAN-Induced Circadian and Metabolic Dysregulation

Multiple studies have documented that chronic exposure to artificial light at night significantly alters sleep architecture and suppresses nocturnal melatonin secretion [29]. This suppression has important metabolic consequences because melatonin is a key regulator of energy homeostasis, modulating processes such as insulin sensitivity, appetite regulation, and lipid metabolism [53].

At the molecular level, melatonin influences hepatic circadian gene expression and metabolic pathways through activation of its receptors MT1 and MT2. ALAN-induced melatonin suppression leads to desynchronization of peripheral clocks, a reduction in the amplitude of rhythmic expression of core clock genes such as Per, Bmal1, and Nocturnin, and impairment of signaling pathways including Phosphoinositide 3-kinase/Protein kinase B (PI3K/Akt) and Mitogen-activated protein kinase/Extracellular signal-regulated kinase (MAPK/ERK). These alterations contribute to the development of insulin resistance and the promotion of hepatic lipogenesis [54].

In addition to its role in circadian regulation, melatonin is a potent endogenous antioxidant that neutralizes reactive oxygen species and modulates the activity of antioxidant enzymes. Deficiency in melatonin due to ALAN exposure increases oxidative stress within the liver, facilitating the progression from simple steatosis to steatohepatitis and hepatocellular damage [55].

Melatonin deficiency also exacerbates peripheral insulin resistance, leading to an increased flux of FFAs to the liver and promoting triglyceride accumulation in hepatocytes—processes central to MASLD pathophysiology [41,53]. This combination of altered metabolic signaling, oxidative stress, and lipid accumulation underscores the multifaceted role of melatonin as both a circadian and metabolic regulator whose disruption may significantly contribute to MASLD development.

### 5.4. Metabolic Pathways Linking ALAN Exposure to Hepatic Lipid Dysregulation and MASLD

Experimental evidence indicates that ALAN exposure promotes hepatic lipid accumulation through multiple mechanisms. One pathway involves the increased uptake of circulating FFAs by the liver, accompanied by enhanced de novo lipogenesis. At the same time, ALAN suppresses lipid synthesis in white adipose tissue and alters circulating levels of adipokines such as leptin and adiponectin. This suggests systemic adipose tissue dysfunction, which undermines its normal role as a regulator of energy homeostasis [56].

ALAN also disrupts the normal circadian rhythmicity of glucose metabolism in the liver. Significant changes have been reported in the expression of hepatic genes such as Glut2, Glucokinase (Gck), and Foxo1, which are critical for glucose uptake, glycolysis, and gluconeogenesis. These alterations contribute to hepatic insulin resistance and an increased rate of gluconeogenesis [56,57].

In addition, ALAN can indirectly impair hepatic lipid metabolism through modulation of the gut microbiota. Experimental models have shown that ALAN exposure induces gut dysbiosis, characterized by altered microbial composition and function. This, in turn, can increase intestinal permeability, promote endotoxin translocation into the portal circulation, and trigger hepatic inflammation. Interestingly, administration of melatonin in these models partially restored the gut microbiota composition and improved metabolic parameters, providing evidence for the existence of a light–gut–liver axis in MASLD development [58].

## 6. Discussion

ALAN refers to any artificial illumination that alters the natural light–dark cycle during nighttime hours, encompassing sources such as streetlights, building exteriors, vehicle headlights, and indoor lighting that spills outdoors. Importantly, light exposure during the few hours preceding an individual’s usual bedtime can markedly influence circadian physiology. Exposure within this pre-sleep window suppresses melatonin secretion, delays its onset, and shifts the internal timing of the circadian clock, thereby affecting sleep initiation and overall rhythm stability. Such exposure—often originating from indoor lighting or electronic devices—can therefore have physiological effects comparable to direct nighttime illumination, particularly when the emitted light is enriched in short wavelengths (blue light, ~460–480 nm).

The growing body of experimental and epidemiological evidence positions ALAN as an underestimated yet significant environmental factor influencing hepatic metabolism. Animal studies have consistently shown that chronic exposure to blue-enriched light (~460–480 nm) alters circadian synchronization at both central and peripheral levels, suppresses melatonin secretion, and disrupts hepatic molecular clocks. These alterations promote insulin resistance, mitochondrial dysfunction, and lipid accumulation within hepatocytes, contributing to the development of MASLD. Moreover, continuous or dim light exposure induces systemic inflammation, adipose tissue dysfunction, and gut dysbiosis, revealing a complex “light–gut–liver” axis with broad metabolic implications.

In humans, observational studies align with these findings, demonstrating associations between nighttime light exposure and increased prevalence of obesity, metabolic syndrome, insulin resistance, and poor sleep quality. These conditions share pathophysiological mechanisms with MASLD, supporting the hypothesis that ALAN acts as a cofactor in disease progression. (Figure 1) However, the absence of longitudinal studies, standardized light quantification, and controlled assessment of circadian disruption limits the establishment of direct causality.

This review integrates evidence from chronobiology, environmental health, and hepatology, offering a comprehensive synthesis that reframes MASLD as not only a metabolic or dietary disorder but also one influenced by modern light environments. It is important to emphasize that the most robust evidence linking circadian disruption to metabolic dysfunction derives from studies in night-shift workers, whose behavioral schedules are persistently misaligned with solar light–dark cycles. By contrast, environmental exposure to artificial light at night in the general population encompasses a wide range of conditions—from low-level residential or outdoor illumination to high-intensity occupational exposure—whose biological impacts differ in magnitude and consistency. Epidemiological studies based on satellite-derived ALAN data should therefore be interpreted with caution, as these large-scale ecological measures capture ambient light levels rather than individual exposure and cannot alone establish causal relationships between exterior lighting and metabolic or hepatic outcomes. Recognizing ALAN as a modifiable environmental risk factor emphasizes the potential of preventive strategies—ranging from circadian-friendly lighting designs to public health campaigns promoting adequate darkness during sleep—to mitigate metabolic and hepatic consequences.

Future review studies should aim to systematically evaluate the heterogeneity in experimental designs and exposure parameters currently used to study ALAN. Particular emphasis should be placed on developing standardized definitions and quantification methods for light exposure, including wavelength composition, intensity, timing, and duration, to allow meaningful comparison across studies. Meta-analytical approaches integrating both animal and human data could help establish dose–response relationships and identify critical exposure thresholds relevant to metabolic and hepatic outcomes. In addition, future reviews should explore interdisciplinary perspectives by incorporating advances from chronobiology, urban lighting engineering, and environmental policy, thereby providing a more holistic framework to guide translational research and evidence-based public health recommendations.

From a clinical and public health perspective, current evidence highlights nighttime artificial light exposure as a potentially modifiable environmental risk factor for metabolic disorders, including MASLD. Mitigation strategies should encompass urban lighting policies that minimize circadian disruption, technological innovations that reduce blue light emissions from LEDs and electronic devices, and sleep hygiene programs that promote darkness during rest. Nevertheless, these conclusions arise from the synthesis of existing observational and experimental studies and therefore warrant further experimental validation. Such investigations are essential to test the causal pathways proposed here, as they may have significant implications for preventive medicine, urban planning, and public health policy.

This study is a narrative review and, as such, is subject to certain inherent limitations. The synthesis relied on previously published experimental and observational studies, which may introduce selection bias and limit reproducibility. Although a comprehensive search strategy was employed across multiple databases, the possibility of omitting relevant studies cannot be fully excluded. Furthermore, variability in experimental designs, methods of light exposure quantification, and outcome measures across studies hinders direct comparison and precludes formal meta-analysis. Future systematic reviews using standardized inclusion criteria and quantitative synthesis are warranted to strengthen the causal interpretation of the observed associations between artificial light at night, circadian disruption, and MASLD.

## 7. Conclusions

Artificial light at night disrupts circadian and metabolic homeostasis, contributing to hepatic steatosis and the pathogenesis of MASLD through melatonin suppression, insulin resistance, and gut–liver dysregulation. Standardizing light exposure assessment and conducting long-term human studies are essential to define safe nocturnal light levels. Reducing exposure to blue-enriched light and promoting darkness during sleep represent practical, low-cost strategies to preserve liver and metabolic health.

## Figures and Tables

**Figure 1 ijerph-22-01729-f001:**
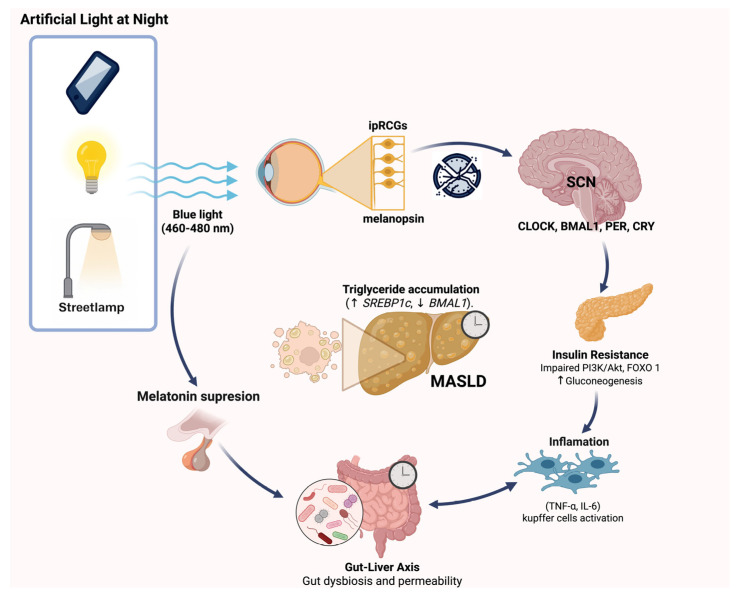
Pathophysiological mechanisms linking artificial light at night (ALAN) exposure to the development of MASLD. Exposure to blue light (460–480 nm) from sources such as mobile devices, indoor lighting, or streetlamps at night activates intrinsically photosensitive retinal ganglion cells (ipRGCs) via melanopsin, disrupting the central circadian clock in the suprachiasmatic nucleus (SCN) and altering the expression of circadian genes (CLOCK, BMAL1, PER, CRY). This leads to insulin resistance through impaired PI3K/Akt signaling and increased gluconeogenesis, as well as inflammation mediated by Kupffer cell activation and pro-inflammatory cytokines (TNF-α, IL-6). Concurrently, ALAN suppresses melatonin secretion, impacting gut-liver axis homeostasis by promoting gut dysbiosis and increased intestinal permeability. These pathways collectively contribute to hepatic triglyceride accumulation (↑SREBP1c, ↓BMAL1), fostering MASLD progression.

## Data Availability

No new data were created or generated in this manuscript. Data sharing is not applicable to this article.

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
