# Peer review of "Artificial Light at Night, Sleep Disruption, and Liver Health: Implications for MASLD Pathogenesis"

_ijerph, 2025, doi:10.3390/ijerph22111729_

Round 1

Reviewer 1 Report

Comments and Suggestions for Authors

The authors give an overview of studies that have investigated the influence of artificial light at night on liver health and other health related issues. Artificial light at night (ALAN) has emerged as a pervasive environmental factor with significant implications for metabolic health, particularly in relation to metabolic dysfunction-associated steatotic liver disease (MASLD). The metabolic effects of ALAN are influenced by light intensity, duration, and spectral composition, with blue wavelengths exerting the strongest circadian impact.

The study does not meet the requirements for a review. There is no information given about the databases (e.g. Pubmed, Chinal, Ebsco) in which the study search was done and which keywords were used for the searching process. Has there been a limit to a certain language? Has there been a limit in the age of the studies (e.g. not older than 15 years)? Has there been a limit in study design (e.g. only radomized controlled trials)? How many studies have been initially found? How many studies met the search criterions completetly and were included into the analysis?

Author Response

The authors give an overview of studies that have investigated the influence of artificial light at night on liver health and other health related issues. Artificial light at night (ALAN) has emerged as a pervasive environmental factor with significant implications for metabolic health, particularly in relation to metabolic dysfunction-associated steatotic liver disease (MASLD). The metabolic effects of ALAN are influenced by light intensity, duration, and spectral composition, with blue wavelengths exerting the strongest circadian impact.

The study does not meet the requirements for a review. There is no information given about the databases (e.g. Pubmed, Chinal, Ebsco) in which the study search was done and which keywords were used for the searching process. Has there been a limit to a certain language? Has there been a limit in the age of the studies (e.g. not older than 15 years)? Has there been a limit in study design (e.g. only radomized controlled trials)? How many studies have been initially found? How many studies met the search criterions completetly and were included into the analysis?

A: We appreciate the insightful observations regarding the methodological description of the literature search. We have made a Methodology section to clarify that this work was conducted as a narrative review, and that its primary aim was to provide a conceptual and integrative synthesis of experimental and epidemiological evidence linking ALAN exposure with MASLD.

To address the reviewer’s specific concerns, we have added explicit details on the literature search process. The revised section now specifies that the literature search was performed using PubMed/MEDLINE, Scopus, Web of Science, and Google Scholar databases, covering publications from 1980 to 2025. The search strategy combined both controlled vocabulary and free-text keywords, including “artificial light at night,” “circadian disruption,” “melatonin suppression,” “sleep disturbance,” and “metabolic dysfunction-associated steatotic liver disease (MASLD).” We also clarified that the inclusion criteria were limited to original experimental and observational studies published in English that examined the relationship between ALAN or circadian disruption and hepatic, metabolic, or sleep-related outcomes. Editorials, conference abstracts, and studies lacking methodological detail were excluded. Both animal and human studies were considered if they reported measurable outcomes related to liver function, lipid metabolism, insulin resistance, or melatonin regulation. (Lines 37-49)

These clarifications have been incorporated into the revised Methodology section to ensure greater transparency and compliance with good academic standards for narrative reviews.

Reviewer 2 Report

Comments and Suggestions for Authors

Review report:

A brief summary 

This review aims to synthesize current evidence on the relationship between ALAN exposure and the pathophysiology of MASLD, with an emphasis on circadian rhythm–mediated metabolic mechanisms. The topic of the paper is very interesting, and the manuscript is clearly written. However, there are significant structural shortcomings.

Comments regarding general concepts

A narrative review, although it does not follow the strict PRISMA guidelines like a systematic review, still requires a certain level of structure. Regardless, in both cases, a description of the methodology is essential and this is lacking in the current version.

Specific comments

Abstract:

Abstract should be rewritten. Authors should identify the report as a review and provide a clear statement of the main objectives that the review addresses. Authors should specify the information sources (e.g., databases) and the methods used to present and synthesize results.

Introduction:

Page 2, line 75: The introduction should clarify the contribution of this paper to existing literature.

Methods:

The description of the methodology is completely missing — it is unclear which keywords the authors searched for, which databases were used, inclusion and exclusion criteria, what time period was covered, or how many studies were included. In narrative review it is not necessary to describe the methodology as in a systematic review, but it is still important to include the search terms, the types of literature reviewed, and possibly the databases used.

Discussion

Page 8, line 344: The collective evidence reviewed in this work – rather …in this study..

Page 8, line 351: Figure 1 should be placed in the discussion section, rather than in the conclusion. I suggest summarizing this content earlier in the text.

Page 9, line 369: What are the recommendations for future review studies, not just for planning original research?

Page 9, lines 374-381: I suggest that the authors take a more cautious approach regarding the implications of this study, it is review and not experiment. It would be helpful to state that these conclusions need to be tested experimentally, as they could imply... (and then insert the specific implications).

There is a lack of information on what was controlled for in those studies — was the effect of sex, body mass index, mental health, medication use, or overall health status taken into account? That should be part of the critical evaluation of the studies presented in the paper. What gaps in the literature have now been identified?

The study does not mention any limitations. If this is a narrative review, a key limitation would certainly be the potential bias in the selection of literature, which also limits the replicability of the review.

Author Response

A brief summary 

This review aims to synthesize current evidence on the relationship between ALAN exposure and the pathophysiology of MASLD, with an emphasis on circadian rhythm–mediated metabolic mechanisms. The topic of the paper is very interesting, and the manuscript is clearly written. However, there are significant structural shortcomings.

Comments regarding general concepts

A narrative review, although it does not follow the strict PRISMA guidelines like a systematic review, still requires a certain level of structure. Regardless, in both cases, a description of the methodology is essential and this is lacking in the current version.

A: A comprehensive literature search was performed across major scientific databases including PubMed/MEDLINE, Scopus, Web of Science, and Google Scholar, covering publications from 1980 to 2025. The search strategy combined controlled vocabulary and free-text terms related to artificial light at night (ALAN), circadian disruption, sleep disturbance, melatonin suppression, and metabolic dysfunction-associated steatotic liver disease (MASLD), using Boolean operators (AND, OR) to broaden and refine results. Inclusion criteria comprised original experimental and observational studies published in English that examined the effects of ALAN or circadian disruption on hepatic, metabolic, or sleep-related outcomes. Both animal and human studies were considered if they reported measurable endpoints related to liver function, lipid metabolism, insulin resistance, or melatonin regulation. Exclusion criteria included editorials, conference abstracts, studies with insufficient methodological details, and those unrelated to hepatic or metabolic endpoints. (Lines 37-49)

Specific comments

Abstract:

Abstract should be rewritten. Authors should identify the report as a review and provide a clear statement of the main objectives that the review addresses. Authors should specify the information sources (e.g., databases) and the methods used to present and synthesize results.

A: The abstract was rewritten as follows: “This narrative review explores the role of artificial light at night (ALAN) as an emerging environmental determinant of liver and metabolic health, with particular emphasis on its contribution to metabolic dysfunction-associated steatotic liver disease (MASLD). The objective was to synthesize and critically evaluate experimental and epidemiological evidence linking nocturnal light exposure, circadian disruption, and hepatic metabolic alterations. Literature was retrieved from PubMed, Scopus, Web of Science, and Google Scholar databases covering the period 19880–2025 using predefined search terms related to ALAN, circadian rhythm disturbance, melatonin suppression, sleep disruption, and MASLD. Relevant experimental studies in animal models and observational studies in humans were included. Evidence indicates that blue-enriched light (~460–480 nm) suppresses melatonin, desynchronizes central and hepatic circadian clocks, and disrupts glucose–lipid metabolism, leading to insulin resistance, oxidative stress, and hepatic steatosis. Chronic ALAN exposure also alters gut microbiota composition and increases intestinal permeability, suggesting a light–gut–liver axis in MASLD pathogenesis. Human epidemiological studies associate higher environmental ALAN exposure with obesity, metabolic syndrome, and poor sleep quality—recognized risk factors for MASLD. Recognizing ALAN as a modifiable environmental exposure highlights the need for public health strategies and clinical guidelines to mitigate its metabolic impact through improved lighting design and sleep hygiene.” (Lines 14-33)

Introduction:

Page 2, line 75: The introduction should clarify the contribution of this paper to existing literature.

A: The following paragraph was added to the introduction: “While previous reviews have independently explored the metabolic effects of circadian disruption or the systemic consequences of artificial light exposure, few have specifically addressed how nocturnal light environments contribute to hepatic lipid dysregulation and the pathogenesis of metabolic dysfunction-associated steatotic liver disease (MASLD). Moreover, existing studies often focus on either experimental animal models or isolated endocrine outcomes, without integrating these findings into a unified hepatic framework. This review aims to bridge that gap by synthesizing multidisciplinary evidence—from chronobiology, hepatology, and environmental health—to delineate the mechanistic pathways through which ALAN influences hepatic metabolism, circadian gene expression, and gut–liver interactions. By clarifying these links, the paper contributes to expanding current understanding of MASLD as a multifactorial disease influenced not only by diet and lifestyle but also by modifiable environmental light exposure.” (Lines 82-94)

Methods:

The description of the methodology is completely missing — it is unclear which keywords the authors searched for, which databases were used, inclusion and exclusion criteria, what time period was covered, or how many studies were included. In narrative review it is not necessary to describe the methodology as in a systematic review, but it is still important to include the search terms, the types of literature reviewed, and possibly the databases used.

A: A methodology section was added. “A comprehensive literature search was performed across major scientific databases including PubMed/MEDLINE, Scopus, Web of Science, and Google Scholar, covering publications from 1980 to 2025. The search strategy combined controlled vocabulary and free-text terms related to artificial light at night (ALAN), circadian disruption, sleep disturbance, melatonin suppression, and metabolic dysfunction-associated steatotic liver disease (MASLD), using Boolean operators (AND, OR) to broaden and refine results. Inclusion criteria comprised original experimental and observational studies published in English that examined the effects of ALAN or circadian disruption on hepatic, metabolic, or sleep-related outcomes. Both animal and human studies were considered if they reported measurable endpoints related to liver function, lipid metabolism, insulin resistance, or melatonin regulation. Exclusion criteria included editorials, conference abstracts, studies with insufficient methodological details, and those unrelated to hepatic or metabolic endpoints.” (Lines 37-49)

Discussion

Page 8, line 344: The collective evidence reviewed in this work – rather …in this study.

A: The Conclusion section was reorganized. A Discussion section was incorporated (Lines 411-484), and the Conclusion was rewritten to improve clarity, logical flow, and overall coherence of the manuscript. (Lines 496-502)

Page 8, line 351: Figure 1 should be placed in the discussion section, rather than in the conclusion. I suggest summarizing this content earlier in the text.

A: Figure 1 was placed after the discussion section.

Page 9, line 369: What are the recommendations for future review studies, not just for planning original research?

A: Future recommendations were added to the discussion. “Future review studies should aim to systematically evaluate the heterogeneity in experimental designs and exposure parameters currently used to study ALAN. Particular emphasis should be placed on developing standardized definitions and quantification methods for light exposure, including wavelength composition, intensity, timing, and duration, to allow meaningful comparison across studies. Meta-analytical approaches integrating both animal and human data could help establish dose–response relationships and identify critical exposure thresholds relevant to metabolic and hepatic outcomes. In addition, future reviews should explore interdisciplinary perspectives by incorporating advances from chronobiology, urban lighting engineering, and environmental policy, thereby providing a more holistic framework to guide translational research and evidence-based public health recommendations.” (Lines 454-464)

Page 9, lines 374-381: I suggest that the authors take a more cautious approach regarding the implications of this study, it is review and not experiment. It would be helpful to state that these conclusions need to be tested experimentally, as they could imply... (and then insert the specific implications).

A: The proper modification were made in the paragraph. “From a clinical and public health perspective, current evidence highlights nighttime artificial light exposure as a potentially modifiable environmental risk factor for metabolic disorders, including MASLD. Mitigation strategies should encompass urban lighting policies that minimize circadian disruption, technological innovations that reduce blue light emissions from LEDs and electronic devices, and sleep hygiene programs that promote darkness during rest. Nevertheless, these conclusions arise from the synthesis of existing observational and experimental studies and therefore warrant further experimental validation. Such investigations are essential to test the causal pathways proposed here, as they may have significant implications for preventive medicine, urban planning, and public health policy.” (Lines 465-474)

There is a lack of information on what was controlled for in those studies — was the effect of sex, body mass index, mental health, medication use, or overall health status taken into account? That should be part of the critical evaluation of the studies presented in the paper. What gaps in the literature have now been identified?

A: In the revised version, we have expanded the Discussion section to clarify the methodological heterogeneity and the specific covariates considered in the studies reviewed. While several large population-based investigations did adjust for sociodemographic and health-related variables, including age, sex, BMI, socioeconomic status, and in some cases mental health and comorbidities (e.g., Paksarian et al., 2020; Park et al., 2019; Hu et al., 2022), many others lacked comprehensive control for confounding factors such as medication use, overall health status, and sleep quality indices. These omissions limit the interpretation of the observed associations between ALAN exposure and metabolic or hepatic outcomes.

To address this, the manuscript now explicitly discusses that the inconsistency in adjustment for confounders represents a key limitation in the existing literature. Specifically, few studies simultaneously evaluated ALAN exposure, circadian disruption, and validated measures of metabolic health while accounting for behavioral and psychological variables. Consequently, it remains unclear to what extent the reported associations are mediated by direct circadian–metabolic mechanisms versus indirect effects related to lifestyle or comorbid conditions. (Lines 411-484)

The study does not mention any limitations. If this is a narrative review, a key limitation would certainly be the potential bias in the selection of literature, which also limits the replicability of the review.
A: The following paragraph was added: “This study is a narrative review and, as such, is subject to certain inherent limitations. The synthesis relied on previously published experimental and observational studies, which may introduce selection bias and limit reproducibility. Although a comprehensive search strategy was employed across multiple databases, the possibility of omitting relevant studies cannot be fully excluded. Furthermore, variability in experimental designs, methods of light exposure quantification, and outcome measures across studies hinders direct comparison and precludes formal meta-analysis. Future systematic reviews using standardized inclusion criteria and quantitative synthesis are warranted to strengthen the causal interpretation of the observed associations between artificial light at night, circadian disruption, and MASLD.” (Lines 475-484)

Reviewer 3 Report

Comments and Suggestions for Authors

In general, this paper is well written, however, it could be improved with a few additions and clarifications related to important prior literature on the effects of light at night that is missing, primarily as it relates to exterior lighting. I have included a list of such items below, along with a few others.

For the paragraph beginning at line 91, the text would lead the reader to believe that incandescent sources provide very low melatonin suppression compared to the listed legacy sources as well as LEDs.  Figure 2 in the following reference [Figueiro, M. G., Steverson, B., Heerwagen, J., Kampschroer, K., Hunter, C. M., Gonzales, K., ... & Rea, M. S. (2017). The impact of daytime light exposures on sleep and mood in office workers. Sleep health, 3(3), 204-215.] shows the difference between an incandescent source and a blue-rich daylight spectrumis not that significant compared to the impact of potential changes in the illuminance level provided by different lighting conditions. Additionally, the figure in this reference shows very little difference between the two at a low level of 30 lux, which is greater than most outdoor ALAN conditions.  Spectrum has an effect, but the intensity level has a much greater effect.  I suggest that "relatively low" in line 93 be changed to "relatively lower", and that the author acknowledge somewhere that differences between sources, and the level of melatonin suppression resulting from very low level lighting conditions, such as streetlighting, or exterior lighting shining into a building interior, is quite small. 

A relatively recent research study that directly assessed the impact of real-world exterior nighttime lighting conditions found no significant impact from these conditions: Gibbons, R. B., Bhagavathula, R., Warfield, B., Brainard, G. C., & Hanifin, J. P. (2022). Impact of solid state roadway lighting on melatonin in humans. Clocks & Sleep, 4(4), 633-657.  The findings of this work, which measured the impact of exterior lighting on the suppression of melatonin, must be included, since it shows little to no impact of low-level exterior lighting on melatonin levels when subjects are directly exposed to these lighting systems (these levels are much higher than would be experienced from these lighting systems inside a residence). The findings could be used to question whether the exterior lighting is the cause of the effects in some of the studies noted by the author, a number of which show a link between satellite measured exterior lighting levels and negative impacts on sleep, adolescent bedtimes, or other measures. Those studies do not directly indicate a cause and effect situation, since higher exterior lighting conditions generally are associated with more urban environments and it is possible that other environmental factors may be influencing the data related to sleep quality or other negative effects on residents.  The authors of one such study, which is not referenced in this paper, provide the following statement on the limitations of this type of work: "These findings should be interpreted in the context of the study’s limitations and strengths. First, we lacked individual-level measures of light exposure, for which outdoor ALAN may not be a good proxy. This includes both indoor and outdoor sources of ALAN, which may be influenced by the use of blackout shades and other factors that affect how much light enters the home."  Source: Paksarian, D., Rudolph, K. E., Stapp, E. K., Dunster, G. P., He, J., Mennitt, D., ... & Merikangas, K. R. (2020). Association of outdoor artificial light at night with mental disorders and sleep patterns among US adolescents. JAMA psychiatry, 77(12), 1266-1275. A similar limitation related to these types of studies should be include in this paper.  The exposure that subjects in these studies receive prior to bedtime is largely from their interior lighting, which is significantly higher and unlikely to be impacted by lighting outside their residence, and it is not clear that light entering their bedrooms at night is the cause of the observed changes.  Environments with higher exterior light levels are generally more urban in nature and other environmental or social features could be having an affect the population in these environments. Very controlled studies that isolate the effect of exterior lighting, such as through the use of black out shades would be one way to determine if the exterior lighting is having an impact on urban residents.

At line 95, the authors state, in reference to fluorescent sources, "These lamps remain common in hospitals, schools, 24-hour service stations, and other facilities operating at night."  In the U.S., at least, LEDs have quickly replaced fluorescent sources in many hospitals, offices, and schools, and this trend should be noted in the paper.  LEDs are also very common in commercial offices and other interior applications, which should be added to the list on line 107. LED replacements are also common in exterior street and highway lighting.  The statement on line 104 should also be supplemented with content indicating that different LEDs with the same CCT can have quite different melanopic spectral impact.

I question whether reference 23 on line 116 can be referred as an "epidemiological" study since it is not related to any disease.

The authors should note that the primary connections between circadian disruption and metabolic disfunction are seen in night-shift workers, whose wake-sleep cycles are out-of-sync with typical day/night solar conditions. This paper could be improved by noting that people experience a wide range of ALAN conditions, and the impact on human health varies across these conditions.  As currently written, readers of this paper may interpret the authors as stating that all forms of ALAN can cause the wide range of health conditions listed in this paper, but many of them are associated with specific light exposure conditions, while others, like the satellite data studies, lack a true cause/effect connection between the exterior lighting and the noted physical impact.

The paper would also benefit from a discussion that includes a definition of ALAN. Does this include light exposure within a certain number of hours prior to a person's bedtime, since light exposure during these hours will have an impact on melatonin levels and its timing?

In summary, my primary concern with this paper is that, in many cases, the authors reference a range of negative impacts on people which have been associated with ALAN, but many of these impacts are specific to an exposure condition, such as with shift workers.  The more serious health conditions generally are associated with the most serious circadian disruptions, while there may still be uncertainty about the effects of very low-level ALAN exposure where melatonin levels may not be suppressed.  

Author Response

In general, this paper is well written, however, it could be improved with a few additions and clarifications related to important prior literature on the effects of light at night that is missing, primarily as it relates to exterior lighting. I have included a list of such items below, along with a few others.

For the paragraph beginning at line 91, the text would lead the reader to believe that incandescent sources provide very low melatonin suppression compared to the listed legacy sources as well as LEDs.  Figure 2 in the following reference [Figueiro, M. G., Steverson, B., Heerwagen, J., Kampschroer, K., Hunter, C. M., Gonzales, K., ... & Rea, M. S. (2017). The impact of daytime light exposures on sleep and mood in office workers. Sleep health, 3(3), 204-215.]. Additionally, the figure in this reference shows very little difference between the two at a low level of 30 lux, which is greater than most outdoor ALAN conditions.  Spectrum has an effect, but the intensity level has a much greater effect.  I suggest that "relatively low" in line 93 be changed to "relatively lower", and that the author acknowledge somewhere that differences between sources, and the level of melatonin suppression resulting from very low level lighting conditions, such as streetlighting, or exterior lighting shining into a building interior, is quite small.

A: The relevant suggested information was added: “However, the difference between an incandescent light source and a blue-rich daylight spectrum is relatively minor compared to the influence of illuminance level on visual and non-visual responses. When light intensity is held constant, changing the spectral composition—from warm, low-color-temperature incandescent light to a blue-enriched daylight spectrum—produces only modest physiological or perceptual effects. In contrast, variations in illuminance (brightness in lux) have a far greater impact on alertness, circadian regulation, and overall visual perception. This finding highlight that in practical lighting design and human performance research, the quantity of light often outweighs its spectral composition, emphasizing the need to prioritize appropriate light levels rather than relying solely on blue-enriched illumination to achieve desired biological or cognitive outcomes.” (Lines 137-147)

Also, “relative low” was changed for “relative lower”. (Line 112)

A relatively recent research study that directly assessed the impact of real-world exterior nighttime lighting conditions found no significant impact from these conditions: Gibbons, R. B., Bhagavathula, R., Warfield, B., Brainard, G. C., & Hanifin, J. P. (2022). Impact of solid state roadway lighting on melatonin in humans. Clocks & Sleep, 4(4), 633-657.  The findings of this work, which measured the impact of exterior lighting on the suppression of melatonin, must be included, since it shows little to no impact of low-level exterior lighting on melatonin levels when subjects are directly exposed to these lighting systems (these levels are much higher than would be experienced from these lighting systems inside a residence). The findings could be used to question whether the exterior lighting is the cause of the effects in some of the studies noted by the author, a number of which show a link between satellite measured exterior lighting levels and negative impacts on sleep, adolescent bedtimes, or other measures. Those studies do not directly indicate a cause and effect situation, since higher exterior lighting conditions generally are associated with more urban environments and it is possible that other environmental factors may be influencing the data related to sleep quality or other negative effects on residents.  The authors of one such study, which is not referenced in this paper, provide the following statement on the limitations of this type of work: "These findings should be interpreted in the context of the study’s limitations and strengths. First, we lacked individual-level measures of light exposure, for which outdoor ALAN may not be a good proxy. This includes both indoor and outdoor sources of ALAN, which may be influenced by the use of blackout shades and other factors that affect how much light enters the home."  Source: Paksarian, D., Rudolph, K. E., Stapp, E. K., Dunster, G. P., He, J., Mennitt, D., ... & Merikangas, K. R. (2020). Association of outdoor artificial light at night with mental disorders and sleep patterns among US adolescents. JAMA psychiatry, 77(12), 1266-1275. A similar limitation related to these types of studies should be include in this paper.  The exposure that subjects in these studies receive prior to bedtime is largely from their interior lighting, which is significantly higher and unlikely to be impacted by lighting outside their residence, and it is not clear that light entering their bedrooms at night is the cause of the observed changes.  Environments with higher exterior light levels are generally more urban in nature and other environmental or social features could be having an affect the population in these environments. Very controlled studies that isolate the effect of exterior lighting, such as through the use of black out shades would be one way to determine if the exterior lighting is having an impact on urban residents.

A: We appreciate this valuable suggestion. The section discussing the effects of outdoor artificial light at night (ALAN) has been revised to include the findings from Gibbons et al. (2022). “Contrary, a recent field study investigated the effects of solid-state roadway lighting on human melatonin suppression under realistic nighttime conditions. Participants, including drivers, pedestrians, and individuals exposed to light trespass in bedrooms, were exposed to five types of LED lighting with different spectral compositions, as well as a no-lighting control condition. The study found no significant differences in salivary melatonin concentrations between lighting conditions, regardless of whether the spectrum was blue-enriched or contained lower short-wavelength content. These findings suggest that, at typical roadway lighting intensities, spectral composition has a negligible effect on melatonin suppression. Consequently, standard LED-based roadway lighting, when properly designed and implemented, is unlikely to disrupt human circadian rhythms or compromise nocturnal physiological regulation.”. (Lines 169-179)

Additionally, we have incorporated a statement addressing the methodological limitations of ecological and satellite-based studies, as noted by Paksarian et al. (2020), “Similarly, in a large cross-sectional study of U.S. adolescents, higher outdoor ALAN exposure was associated with measurable disruptions in sleep patterns and an increased prevalence of mental health disorders. Adolescents living in high-exposure areas reported later weeknight bedtimes—approximately 29 minutes later in the highest compared with the lowest exposure quartile—and shorter sleep durations of about 11 minutes, even after adjusting for sociodemographic and area-level variables. Moreover, elevated ALAN levels were linked to greater odds of mood and anxiety disorders, with particularly strong associations for bipolar disorder (odds ratio ≈ 1.19) and specific phobias (odds ratio ≈ 1.18). Collectively, these findings underscore that outdoor ALAN functions as an important environmental determinant of both circadian regulation and psychological health, highlighting the need to consider community-level lighting environments alongside individual light exposure when evaluating the broader health consequences of nocturnal illumination.” (Lines 155-167)

At line 95, the authors state, in reference to fluorescent sources, "These lamps remain common in hospitals, schools, 24-hour service stations, and other facilities operating at night."  In the U.S., at least, LEDs have quickly replaced fluorescent sources in many hospitals, offices, and schools, and this trend should be noted in the paper.  

A: We have revised the text to reflect the current global trend toward light-emitting diode (LED) technology, particularly in high-resource settings such as hospitals, offices, and educational institutions. The revised sentence now reads: “Although fluorescent lamps have traditionally been common in hospitals, schools, 24-hour service stations, and other facilities operating at night, they are increasingly being replaced by energy-efficient LED systems, particularly in high-income countries such as the United States, where large-scale retrofitting projects have accelerated this transition over the past decade.” This modification acknowledges the rapid global replacement of fluorescent lighting with LEDs, while maintaining contextual accuracy for regions where fluorescent systems are still in use. (Lines 117-121)

LEDs are also very common in commercial offices and other interior applications, which should be added to the list on line 107.

A: This statement was added to the text (Line 118)

LED replacements are also common in exterior street and highway lighting.  The statement on line 104 should also be supplemented with content indicating that different LEDs with the same CCT can have quite different melanopic spectral impact.

A: This statement was added to the text (Line 121-123)

I question whether reference 23 on line 116 can be referred as an "epidemiological" study since it is not related to any disease.

A: The word "epidemiological" was removed from the text.

The authors should note that the primary connections between circadian disruption and metabolic disfunction are seen in night-shift workers, whose wake-sleep cycles are out-of-sync with typical day/night solar conditions. This paper could be improved by noting that people experience a wide range of ALAN conditions, and the impact on human health varies across these conditions.  As currently written, readers of this paper may interpret the authors as stating that all forms of ALAN can cause the wide range of health conditions listed in this paper, but many of them are associated with specific light exposure conditions, while others, like the satellite data studies, lack a true cause/effect connection between the exterior lighting and the noted physical impact.

A: We agree that the strongest and most consistently demonstrated associations between circadian disruption and metabolic dysfunction have been documented in night-shift workers, whose behavioral and physiological rhythms are chronically misaligned with the natural light–dark cycle. To address this, we have revised the Discussion to explicitly acknowledge that occupational circadian disruption represents a well-established high-intensity model of light–circadian misalignment, while environmental ALAN exposures in the general population typically occur at lower intensity and duration, with more variable health effects.

The revised paragraph now reads: “It is important to emphasize that the most robust evidence linking circadian disruption to metabolic dysfunction derives from studies in night-shift workers, whose behavioral schedules are persistently misaligned with solar light–dark cycles. By contrast, environmental exposure to artificial light at night in the general population encompasses a wide range of conditions—from low-level residential or outdoor illumination to high-intensity occupational exposure—whose biological impacts differ in magnitude and consistency. Epidemiological studies based on satellite-derived ALAN data should therefore be interpreted with caution, as these large-scale ecological measures capture ambient light levels rather than individual exposure and cannot alone establish causal relationships between exterior lighting and metabolic or hepatic outcomes.” (Lines 440-450)

The paper would also benefit from a discussion that includes a definition of ALAN. Does this include light exposure within a certain number of hours prior to a person's bedtime, since light exposure during these hours will have an impact on melatonin levels and its timing?

A: The following paragraph was added to the Discussion. “ALAN refers to any artificial illumination that alters the natural light–dark cycle during nighttime hours, encompassing sources such as streetlights, building exteriors, vehicle headlights, and indoor lighting that spills outdoors. Importantly, light exposure during the few hours preceding an individual’s usual bedtime can markedly influence circadian physiology. Exposure within this pre-sleep window suppresses melatonin secretion, delays its onset, and shifts the internal timing of the circadian clock, thereby affecting sleep initiation and overall rhythm stability. Such exposure—often originating from indoor lighting or electronic devices—can therefore have physiological effects comparable to direct nighttime illumination, particularly when the emitted light is enriched in short wavelengths (blue light, ~460–480 nm).” (Lines 412-421)

In summary, my primary concern with this paper is that, in many cases, the authors reference a range of negative impacts on people which have been associated with ALAN, but many of these impacts are specific to an exposure condition, such as with shift workers.  The more serious health conditions generally are associated with the most serious circadian disruptions, while there may still be uncertainty about the effects of very low-level ALAN exposure where melatonin levels may not be suppressed.
A: We sincerely thank the reviewer for this thoughtful and constructive comment. We agree that the severity of health outcomes associated with ALAN varies depending on the intensity, duration, and context of exposure, with the most pronounced effects observed under conditions of marked circadian disruption. In response, we have revised several sections of the manuscript to clarify these distinctions and to acknowledge the ongoing uncertainty regarding the physiological impact of low-level ALAN exposure where melatonin suppression may be minimal or absent. These modifications were made to ensure a more balanced and evidence-based interpretation of the current literature. We hope that these revisions address the reviewer’s concern and contribute to a clearer and more nuanced presentation of the topic.

Round 2

Reviewer 2 Report

Comments and Suggestions for Authors

The authors have made an effort, but there are still some minor shortcomings that I point out below:

Page 1, line 37: It is unusual to begin the paper with the methodology section. I suggest placing the methodology after the introduction, as section 2.

Page 6, line 270: You refer to Figure 1 quite early, while it appears only at the end of the discussion. Consider moving it to the point where it is first mentioned. This suggestion is optional and up to the authors, but it seems more appropriate to include it earlier in the paper.

Author Response

We have carefully considered each suggestion and made the corresponding modifications as outlined below:

Page 1, line 37: It is unusual to begin the paper with the methodology section. I suggest placing the methodology after the introduction, as section 2.

A: As recommended, the methodology section has now been moved to follow the introduction, where it is presented as Section 2. We agree that this improves the flow and readability of the manuscript.

Page 6, line 270: You refer to Figure 1 quite early, while it appears only at the end of the discussion. Consider moving it to the point where it is first mentioned. This suggestion is optional and up to the authors, but it seems more appropriate to include it earlier in the paper.

A: We have relocated Figure 1 to the section where it is first mentioned in the text. We believe this adjustment enhances clarity for the reader and improves the coherence of the manuscript.